# Ex Vivo Assessment of Tumor-Targeting Fluorescent Tracers for Image-Guided Surgery

**DOI:** 10.3390/cancers12040987

**Published:** 2020-04-17

**Authors:** Fortuné M.K. Elekonawo, Jan Marie de Gooyer, Desirée L. Bos, David M. Goldenberg, Otto C. Boerman, Lodewijk A.A. Brosens, Andreas J.A. Bremers, Johannes H.W. de Wilt, Mark Rijpkema

**Affiliations:** 1Department of Radiology, Nuclear Medicine and Anatomy, Radboud Institute for Health Sciences, Radboud University Medical Center, 6525GA Nijmegen, The Netherlands; 2Department of Surgery, Radboud University Medical Center, 6525GA Nijmegen, The Netherlands; 3Center for Molecular Medicine and Immunology, Mendham, NJ 07945, USA; 4Department of Pathology, Radboud University Medical Center, 6525GA Nijmegen, The Netherlands

**Keywords:** image-guided surgery, colorectal cancer, translational medicine, targeted imaging near infrared fluorescence, antibody, ex vivo

## Abstract

Image-guided surgery can aid in achieving complete tumor resection. The development and assessment of tumor-targeted imaging probes for near-infrared fluorescence image-guided surgery relies mainly on preclinical models, but the translation to clinical use remains challenging. In the current study, we introduce and evaluate the application of a dual-labelled tumor-targeting antibody for ex vivo incubation of freshly resected human tumor specimens and assessed the tumor-to-adjacent tissue ratio of the detectable signals. Immediately after surgical resection, peritoneal tumors of colorectal origin were placed in cold medium. Subsequently, tumors were incubated with ^111^In-DOTA-hMN-14-IRDye800CW, an anti-carcinoembryonic antigen (CEA) antibody with a fluorescent and radioactive label. Tumors were then washed, fixed, and analyzed for the presence and location of tumor cells, CEA expression, fluorescence, and radioactivity. Twenty-six of 29 tumor samples obtained from 10 patients contained malignant cells. Overall, fluorescence intensity was higher in tumor areas compared to adjacent non-tumor tissue parts (*p* < 0.001). The average fluorescence tumor-to-background ratio was 11.8 ± 9.1:1. A similar ratio was found in the autoradiographic analyses. Incubation with a non-specific control antibody confirmed that tumor targeting of our tracer was CEA-specific. Our results demonstrate the feasibility of this tracer for multimodal image-guided surgery. Furthermore, this ex vivo incubation method may help to bridge the gap between preclinical research and clinical application of new agents for radioactive, near infrared fluorescence or multimodal imaging studies.

## 1. Introduction

Successful surgical treatment of many cancers relies on complete tumor resection; incomplete resection increases the odds of recurrence [1,2,3,4,5,6,7]. Especially in late-stage cancers, complete resection of tumors and accurate detection of their metastases remains challenging. Intraoperative imaging techniques, such as real-time near-infrared fluorescence imaging, might help to overcome these challenges [8]. Currently, several clinical trials in patients with various types of cancer are ongoing, and others have demonstrated that fluorescent imaging probes can be used for intraoperative or postoperative evaluation of tumor margins and the presence of cancer [9,10]. Tumor-specific fluorescent or multimodal tracers might help to increase the surgeons’ ability to discriminate between healthy and malignant tissue, and several clinical trials in different cancers evaluated their use [8]. 

There are several prerequisites for successful tumor-targeted fluorescent image-guided surgery. Obviously, the amount of tracer accumulating in the tumor should surpass the detection limit of the fluorescence imaging camera system. Second, the fluorescent tracer should reach sufficiently high tumor-to-background tissue ratios (TBRs) in vivo to discriminate between tumor and adjacent normal tissue. 

Newly developed fluorescent tracers for use in surgical oncology are initially tested in vitro for their tumor-targeting potential, using library-derived cancer cell lines that are cultured under standardized laboratory conditions. Subsequent steps may include small animals, (patient-derived) organoid models, or the ex vivo perfusion of organs. Animal models can provide information on biodistribution, clearance, and toxicity. However, the ability of preclinical results to predict clinical practice is limited [11]. For fluorescent tracers this applies to the TBR found in patients. 

Therefore, we present a method to assess the TBR for predicting the clinical feasibility of a fluorescently labeled tracer for image-guided surgery. In short, we incubated freshly resected peritoneal tumors of patients with colorectal peritoneal carcinomatosis with a multimodal anti-carcinoembryonic antigen (CEA) antibody (^111^In-DOTA-hMN-14-IRDye800CW), because CEA is overexpressed in colorectal cancer and can be used as a highly specific target for fluorescently labeled tracers [12,13]. The feasibility of this multimodal antibody for fluorescence image-guided surgery and detection of metastasis has already been demonstrated in mice with human tumor xenografts [14,15]. The aim of this study was to determine the tumor to adjacent tissue contrast after ex vivo tissue incubation with ^111^In-DOTA-hMN-14-IRDye800CW to assess the feasibility of using this tracer for fluorescence image-guided surgery in cancer patients.

## 2. Results

We collected 29 peritoneal deposits from 10 colorectal cancer patients (5 males, 5 females). Mean age was 64.8 ± 10.7 years. Four patients received adjuvant systemic therapy for their primary tumor prior to cytoreductive surgery and hyperthermic intraperitoneal chemotherapy (CRS-HIPEC). Six patients were diagnosed with a peritoneally metastasized adenocarcinoma, and in three patients the peritoneal metastases originated from a mucinous adenocarcinoma. In one patient, the metastases originated from signet-ring-cell carcinoma.

After incubation, tissue remained microscopically viable as was determined microscopically on H&E slides by the involved pathologist. Twenty-six tumors contained malignant cells, and depending on size and quality, multiple or single series of slides were produced from these tumors. This resulted in 46 series of slides that were analyzed. The other three tumors did not contain viable tumor cells but mainly consisted of fibrosis or stroma. Median fluorescence intensity was 453.4 (IQR: 257.2–919.0) in tumors, whereas in adjacent tissue the median fluorescence intensity was 48.7 (IQR: 28.8–79.0) (*p* < 0.001). Overall, fluorescence intensity was higher in tumorous areas compared to adjacent non-tumor tissue parts (Figure 1). Mean fluorescence intensity in tumor tissue did not differ among patients with or without a history of systemic therapy (*p* = 0.912). Median intensity of the autoradiography for tumor tissue was 5.0∙10^6^ (IQR: 2.4∙10^6^–9.2∙10^6^), while the median autoradiography intensity in non-tumor tissue was 9.9∙10^5^ (IQR: 2.5∙10^5^–2.4∙10^6^) (*p* < 0.001). The TBRs for the fluorescence and radio signal in each patient is shown in Appendix A. An example of a tumor and normal tissue ROI is provided in Figure 2.

Tumors of two patients were incubated with dual-labeled hMN-14 (^111^In-DOTA-hMN-14-IRDye800CW) in parallel with dual-labeled hIgG as control (Figure 1; last 2 patients). Median tumor fluorescence intensity of hIgG treated samples was 4.9 (IQR 2.7–8.5) which was similar to the fluorescence intensity of normal tissue in the same samples: 4.9 (IQR 3.6–13.3, *p* = 0.602). Similarly, the median intensity of the autoradiography was 5.6∙10^5^ (IQR: 4.5∙10^5^–7.5∙10^5^) for tumor tissue and 4.4∙10^5^ (IQR: 3.8∙10^5^–7.5∙10^5^) for non-tumorous tissue (*p* = 0.465). Furthermore, in the in vitro binding assay (Appendix A), dual-labeled hMN-14 showed higher binding to LS147T cells than the non-specific hIgG conjugate (*p* < 0.001). Additional blocking with an excess of unlabeled antibody led to a significant reduction in binding (*p* < 0.001), indicating specific binding of ^111^In-DOTA-hMN-14-IRDye800CW to CEA (Appendix A).

## 3. Discussion

We observed high tumor-to-surrounding tissue ratios of our dual anti-CEA tracer ^111^In-DOTA-hMN-14-IRdye800CW after ex vivo incubation of freshly resected colorectal peritoneal metastases. Together with earlier results on biodistribution and tumor accumulation, these results indicate that it is feasible to use this tracer for fluorescence image-guided surgery in patients with colorectal peritoneal metastases. This way, ex vivo incubation of surgical samples contributes to bridging the gap between preclinical studies and clinical application of novel tracers for fluorescence and multimodal image-guided surgery.

Fluorescent and radiolabeled bimodal imaging probes may serve a versatile role before, during, and after image-guided surgery. This includes accurate tracer quantification for pharmacokinetic purposes, preoperative radionuclide imaging, real-time intraoperative radiation detection, real-time near-infrared fluorescent imaging, and qualitative and quantitative ex vivo analysis of resection specimens as has been demonstrated in several translational studies for multiple diseases [15,16,17,18]. Furthermore, its feasibility has been demonstrated in recent clinical trials [19,20], and several clinical trials are currently ongoing [8]. Ex vivo incubation of patient tissue specimens with antibodies has previously been performed in different applications [21,22]. In the present study, we applied this approach to assess the TBR of multimodal antibody conjugates to be used for image-guided surgery. The involved pathologist assessed all included tumor specimens after incubation, and they remained viable based on microscopic H&E assessment. However, the incubated tissue could undergo molecular changes that may not be visible on H&E stainings. These were not assessed in the current study.

Since clinical translation of preclinical results is usually challenging, we incubated freshly resected human tumor specimens with the multimodal tracer in-vitro. We chose an incubation concentration similar to the blood concentration that is to be expected following intravenous injection of approximately 20 mg antibody-conjugate in a patient. This dose is similar to doses used in clinical image-guided surgery studies with tumor-targeting antibody conjugates [23,24]. We evaluated the potential of our tracer using colorectal peritoneal metastases from a group of patients receiving surgical treatment. On average, we found an almost ten-fold higher microscopic fluorescent signal in CEA-positive tumor areas compared with adjacent non-tumorous areas in all samples after incubation with dual-labelled hMN-14. Moreover, the fluorescent signal in cancer cells was at least 5-fold higher than the signal in adjacent normal tissue in all resected tumor specimens (Figure 1), indicating sufficient tumor-to-background contrast in all patients. We found a similar contrast in the autoradiographic analyses. Recent clinical studies describe TBRs between two and five after intravenous injection of fluorescently or dual-labeled tumor-targeting antibodies [24,25,26,27]. Methodological differences in analyses or tumor type might explain the difference in TBR between literature and our findings. Moreover, the main limitation of ex vivo incubation of surgical specimens is that it does not represent the clinical administration route (intravenous; circulation) and the (slow) clearance in the clinical situation. Consequently, tracer accumulation is not limited by extravasation or plasma clearance. Instead, it relies on passive diffusion of the antibody conjugate into tissue and release of unbound tracer into the washing buffer during the washing step. We noticed less penetration to CEA-positive areas when we analyzed deeper than 100 µm in incubated specimens. Longer incubation time or a smaller tumor-targeting molecule might result in deeper tissue penetration. However, in general during surgery tumor edges and margins are of more interest than tumor cores. Therefore, analyzing only the superficial layers after ex vivo incubation might be sufficient to assess the feasibility of a tracer for image-guided surgery (this may be different for therapeutic applications). 

A key factor determining the TBR of tumor-targeting tracers is the presence and availability of the biomarker that the tracer targets. All evaluated tumor specimens were found positive for CEA. In vivo, membrane-bound CEA can shed into the interstitial compartment and circulation [28]. As a result, our tracer could bind to soluble serum CEA after shedding. An earlier study showed that the targeting sensitivity of MN-14 is not affected by complexation with plasma CEA, and that complexation reduces at increasing doses [29]. More recently, Boogerd et al. [26] used an anti-CEA tracer in a clinical study with 26 patients and observed that only a minor fraction (3%) of the administered dose was lost by binding to circulating CEA. Due to the nature of our ex vivo incubation assay, investigation of CEA present in the circulation is not possible. However, we did observe that our tracer accumulated in CEA-positive mucin (Appendix A). Primary mucinous adenocarcinomas are more often associated with positive resection margins [30]. Since viable tumor cells may reside in or between pools of mucin, fluorescence imaging might also be of added value during surgery of this subtype of colorectal cancer. In contrast to mucin, necrotic areas showed less uptake of our tracer compared to viable tumor cells (Appendix A), indicating high specificity of our tracer for CEA-expressing tumor cells. 

Another method to more closely mimic the clinical situation is ex vivo perfusion of organs or tumors. There, the biological architecture, including blood vessels, remains largely preserved [31]. Unfortunately, ex vivo perfusion studies are limited by the need of afferent arteries that can be cannulated and connected to a perfusion system. When tumors are small and vascular architecture does not allow for such connection, ex vivo incubation can be an alternative method. This way, tissue integrity remains intact even when a (macro)vascular structure is missing. Recent advances also indicate that patient-derived organoids can be used to predict patient-specific drug responses [32]. While developments in coculturing the tumor immune-microenvironment are promising [33], creating similar stroma compositions including the presence of necrosis, immune cells or other non-malignant components of the tumor remains challenging in tumor models. Therefore, the method presented in this study offers specific advantages for translational evaluation of tracers for imaging, especially when tumors of interest are small. Furthermore, it may contribute to bridging the gap between translational research and clinical application of new agents for multimodal or near-infrared fluorescence imaging studies. We showed that high TBRs can be found after ex vivo tissue incubation with dual-labeled hMN-14. These promising results would support further clinical studies using this tracer in patients with peritoneal metastases of colorectal origin. 

## 4. Materials and Methods 

### 4.1. Ethical Approval

All experiments have been carried out according to the principles of the Declaration of Helsinki. The study protocol and collection of tissue were approved by the research ethics committee of the region Arnhem-Nijmegen (CMO regio Arnhem-Nijmegen) on 01-05-2017 (ethic code: 1004-2017).

### 4.2. Patients and Patient Tissue Samples

We collected tissue from 11 patients with peritoneal metastases of colorectal origin who underwent cytoreductive surgery combined with hyperthermic intraperitoneal chemotherapy (CRS-HIPEC). According to the approved protocol, only surgical resection specimens of tumors that did not need standard pathological evaluation were collected (i.e., tissue surplus). Patient data were anonymized for members of the research team that did not have a direct treatment relationship with the patients. Therefore, no informed consent was required as approved by the ethics committee. One patient was excluded because the processing time (resection to incubation) was too long (>2 h) which led to tissue of inferior viable quality.

### 4.3. Antibody Preparation

The hMN-14 (labetuzumab) was kindly provided by Immunomedics, Inc. (Morris Plains, NJ, USA). Labetuzumab is a humanized IgG directed against carcinoembryonic antigen-related cell adhesion molecule 5 (CEACAM5) with high affinity [34]. We used a human isotype IgG (hIgG) as a negative control: human IgG-UNLB (SouthernBiotech, Birmingham, AL, USA). Prior to conjugation, stock hIgG was dialyzed in a Slide-A-Lyzer (10 kDa cutoff; Thermo Fisher Scientific, Waltham, MA, USA) against phosphate-buffered saline (PBS). For the current experiments, hMN-14 and hIgG were conjugated with IRDye^®^ 800CW-NHS (LI-COR, Lincoln, NE, USA) and p-SCN-Bn-DOTA (Macrocyclics, Plano, TX, USA) in two steps. First, hMN-14 (9.6 mg/mL) and hIgG (0.63 mg/mL) were conjugated with IRDye800CW-NHS in 0.1 M NaHCO_3_, pH 8.5, with a 3 fold molar excess of IRDye800CW-NHS. Next, the reaction mixture was incubated for 1 h at room temperature on an orbital shaker and protected from light. Second, DOTA-NHS in 0.1 M NaHCO_3_, pH 9.5, was added to the reaction mixture in a 10-(hMN-14) and 20 fold (hIgG) molar excess. After another hour of incubation on the orbital shaker in the dark, the mixture was dialyzed in a Slide-A-Lyzer (10 kDa cutoff; Thermo Fisher Scientific, Waltham, MA, USA) against 0.25 ammonium acetate (NH_4_Ac), pH 5.5, containing 2 g/L Chelex^®^ 100 Resin (Bio-Rad Laboratories, Inc., Hercules, CA, USA). The final concentration of the conjugates was determined spectrophotometrically using the Infinite 200^®^ Pro (Tecan group Ltd., Männedorf, Switzerland) measuring at 280 nm, correcting for the absorption of IRDye800CW. The substitution ratio of the dye reached 1.09 for hMN-14 and 0.88 for hIgG.

### 4.4. Radiolabeling of the Antibody Conjugates

Briefly, [^111^In]InCl_3_ (Curium, Petten, The Netherlands) was added to dual-conjugated hIgG or hMN-14 in two volumes of 0.1 M 2-(N-morpholino)ethanesulfonic acid (MES), pH 5.5. After 45 min of incubation at 40 °C, 50 mM ethylenediaminetetraacetic acid (EDTA) was added to the labeling reaction in a final concentration of 5 mM to chelate unincorporated [^111^In]InCl_3_. Labeling efficiency was determined by instant thin-layer chromatography (ITLC) on Varian silica gel strips (ITLC-SG; Agilent Technologies, Amstelveen, The Netherlands), using 0.1 mM ammonium acetate (NH_4_Ac) buffer with 0.1 M EDTA, pH 5.5, as the mobile phase. Antibody-conjugates were purified by gel filtration on a PD-10 column, and the radiochemical purity of the final dual-labeled conjugates reached >95% as determined by ITLC.

### 4.5. In Vitro Antibody Testing

To confirm that the hMN-14 antibody-conjugate was immunoreactive for CEA and the non-specific control hIgG conjugate was not, the two antibody preparations were tested in an in vitro binding assay. The DOTA-hIgG-IRDye800CW and DOTA-hMN-14-IRDye800CW were radiolabeled with 0.04 MBq/µg of [^111^In]InCl_3_. 

Briefly, the radiolabeled conjugates were diluted to contain 400 Bq/µL. To determine the amount of activity, we used a shielded 3”-well-type γ-counter (PerkinElmer, Boston, MA, USA). Mycoplasma-negative, human CEA-positive colon adenocarcinoma LS174T cells (ATCC, Manassas, VA, USA) were cultured according the supplier’s instructions without antibiotic additive. Cells were counted and placed in RPMI 1640 medium (ThermoFisher Scientific, Waltham, MA, USA) supplemented with 0.5% bovine serum albumin (BSA; Sigma–Aldrich Chemie N.V., Zwijndrecht, The Netherlands) (binding buffer; BB). Next, cells were pipetted in 12 1.5 mL Eppendorf tubes (1.2·10^6^ cells per tube). We added 40 kBq of the respective radiolabeled antibody conjugate (hMN-14 or hIgG) to cells in each tube (6 tubes with hMN-14; 6 with hIgG). An excess of unlabeled DOTA-hMN-14-IRDye800CW (>2.0 µg) was added to 3 tubes of each antibody conjugate to determine the amount of nonspecific binding of the conjugate. Cells were incubated at 37 °C with 5% CO_2_ for 4 h. Following incubation, cells were centrifuged at 800× *g* for 5 min, the supernatant was removed, and the remaining activity in the tubes was measured in the γ-counter. A 100 µL standard, representing 100% activity, was measured in triplicate simultaneously. 

Analyses of the fluorescence signal of the antibody conjugate were performed as described by Rijpkema et al. [15].

### 4.6. Patient Tissue Incubation and Handling

Immediately after resection, specimens were placed in 4–8 °C low glucose Dulbecco’s modified Eagle’s medium (DMEM) (ThermoFisher Scientific, Waltham, MA, USA), supplemented with penicillin–streptomycin (Gibco^TM^, ThermoFisher Scientific, Waltham, MA, USA) (10,000 U/mL penicillin and 10,000 µg/mL streptomycin), diluted in DMEM to a final concentration of 100 U/mL (Figure 3A). Within 1 h after resection, adipose tissue parts were discarded and, to promote tissue penetration of the antibody-conjugate, several larger (>3 mm) individual tumors were divided into 2 or more similar parts by sharp dissection (maximum 2 × 2 mm). Subsequently, tumors were placed in DMEM supplemented with 100 U/mL penicillin–streptomycin and 0.1% BSA, and 4 µg∙mL^−1^, indium-111-labeled antibody conjugate. The incubation concentration was similar to the blood concentration that is to be expected directly following intravenous injection of approximately 20 mg antibody-conjugate in a patient with 5 L of blood. Clinical image-guided surgery studies with tumor-targeting antibody conjugates used similar doses [23,24]. Tumors were incubated in 5–10 mL of incubation medium on an orbital shaker at 37 °C in an atmosphere containing 5% CO_2_ (Figure 3B). Incubation was performed overnight (mean incubation time: 20 h ± 1 h). Following incubation, tumors were washed in at least 4.5 L of continuously moving and serially refreshed PBS supplemented with 0.1% BSA at 4–8 °C for a minimum of 4 h (Figure 3C).

Specimens were fixed in 4% formaldehyde and embedded in paraffin. After embedding, 4 µm histological sections were cut to a maximum of 150 µm from the surface. Sections were mounted on immunohistochemical microscope glass slides. Four serial slides were used for fluorescence imaging, macroautoradiography, H&E staining, and immunohistochemistry (IHC) (Figure 3D). The involved pathologist (LB) microscopically evaluated on H&E slides if the tissue was suitable for histopathological analyses after the incubation process. The IHC was performed to evaluate carcinoembryonic antigen (CEA), cytokeratin 20 (CK20), and homeobox protein 2 (CDX-2). The slide designated for H&E staining was first imaged on a closed field fluorescence imaging system (Odyssey CLx; LI-COR Biosciences, Lincoln, NE, USA). Next, the same slide was placed on a photostimulable phosphor plate. After 10 days, we imaged the plate using photo-stimulated luminescence on a phosphor imager (Typhoon FLA 7000 phosphor imager, GE Healthcare, Hoevelaken, The Netherlands). Thereafter, standard H&E staining was performed according to local protocol. 

We included two patients for the hIgG antibody-conjugate control. In both patients, the tumors were cut into 2 equal halves so that the same tumor could be incubated with both antibody conjugates (^111^In-labeled DOTA-hMN-14-IRDye800CW and ^111^In-labeled DOTA-hIgG-IRDye800CW).

### 4.7. Immunohistochemistry

Slides were deparaffinized with xylene, rehydrated in ethanol, and rinsed in distilled water according to standard local protocol. Heat-induced antigen retrieval was performed in EDTA solution (pH 9.0). Endogenous peroxidase activity was blocked with 3f H_2_O_2_ in 10 mM PBS for 10 min at room temperature. Subsequently, tissue sections were washed with 10 mM PBS and stained with primary antibodies (CEA: mouse monoclonal; Clone COL-1; MS-613-P; Neomarkers Fremont, CA, USA. CK20: rabbit Monoclonal; Clone E19-1; VWRKILM2133-C1; ImmunoLogic/VWR International B.V., Amsterdam, The Netherlands. CDX2: rabbit monoclonal; Clone EPR2764Y; 235R-16; Cell Marque/Millipore Sigma, Darmstadt, Germany).

Next, sections were incubated with biotin free Poly-HRP-anti mouse/rabbit IgG (ImmunoLogic/VWR International B.V., Amsterdam, The Netherlands) in EnVision™ FLEX Wash Buffer (Dako Denmark A/S, Glostrup, Denmark) (1:1) at room temperature for 30 min. Antibody binding was visualized using the EnVision™ FLEX Working Solution (Dako A/S) at room temperature for 10 min. All sections were counterstained with hematoxylin for 5 s before dehydration in ethanol and cover slipping.

### 4.8. Image Analyses

An experienced board-certified gastrointestinal pathologist (LB) reviewed the H&E, CEA, CK20, and CDX2 slides of all resected specimens and verified the location and presence of tumor cells. 

Based on H&E, regions of interest (ROIs) (tumor and adjacent normal) were drawn in the corresponding fluorescent images using Image Studio Lite version 5.2.5 (LI-COR Biosciences, Lincoln, NE, USA). Fluorescence intensity was calculated as provided by the program: mean fluorescence intensity per pixel (fluorescence intensity) which is an arbitrary unit. Background intensity (zero signal) was determined in an area in close proximity to the analyzed tissue and was automatically subtracted from the ROIs using background subtraction in Image Studio Lite. The total tumor fluorescence intensity was divided by the total area of the tumor ROI. The total normal intensity was divided by the total area of the normal ROI. 

The same principle, including automatic background subtraction, was adapted for analysis of the images acquired with photo stimulated luminescence. Here, we used Aida Image Analyzer version 4.21 (Elysia-Raytest, Angleur, Belgium) which calculates a mean pixel intensity.

### 4.9. Statistical Analyses

Statistical analyses were performed with the Statistical Package for Social Sciences, Version 22.0 (IBM Corp., Armonk, NY, USA). All tests were performed two-sided and a significance level of <0.05 was considered to be statistically significant. To test for difference in tumor and normal fluorescence and autoradiography intensities, we performed a Mann–Whitney U test across all samples or control antibody-conjugate-incubated samples. For analyses of patients who received systemic therapy versus patients who did not, we also performed a Mann–Whitney U test which divided the cases by history of systemic therapy.

## 5. Conclusions

The method presented in this study offers specific advantages for translational evaluation of tracers for imaging, especially when tumors of interest are small. Furthermore, it may contribute to bridging the gap between translational research and clinical application of new agents for multimodal or near-infrared fluorescence imaging studies. We showed that high TBRs can be found after ex vivo tissue incubation with dual-labeled hMN-14. These promising results would support further clinical studies using this tracer in patients with peritoneal metastases of colorectal origin. 

## 6. Patents

David M. Goldenberg is retired chairman and founder of Immunomedics, Inc., and IBC Pharmaceuticals, Inc., and holds royalty-bearing patents.

## Figures and Tables

**Figure 1 cancers-12-00987-f001:**
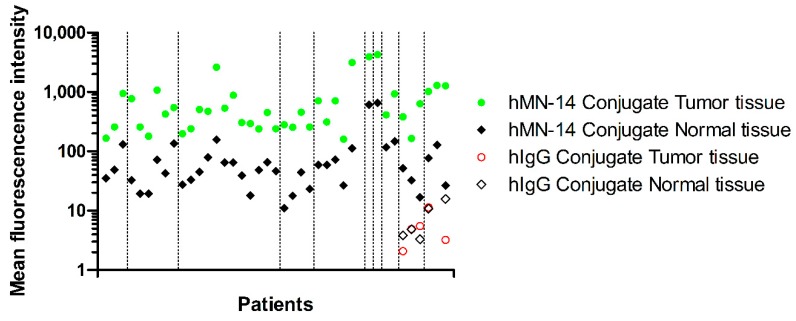
Mean fluorescence intensity (arbitrary units) per pixel for tumor (green dots) and normal tissue (black diamonds) in individual tumors. Each green circle represents an included tumor. Vertical dashed lines separate patients. Note the higher fluorescence signal in all tumors compared to surrounding normal tissue (*p* < 0.001). The control condition (incubation with the non-specific antibody-conjugate DOTA-hIgG-IRDye800CW) shows no significant difference between tumor and normal tissue tracer accumulation (red circles and black open diamond; last two patients).

**Figure 2 cancers-12-00987-f002:**
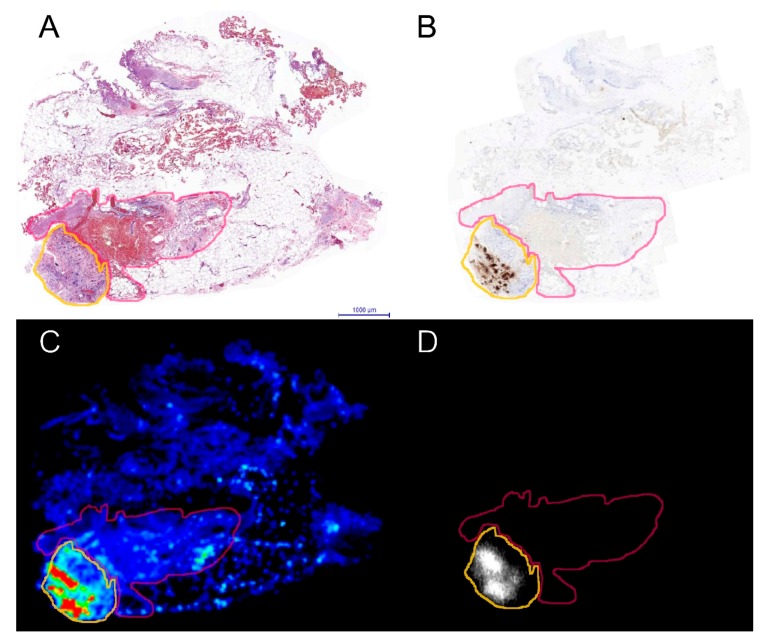
Example of an ROI for tumor (orange line) and surrounding tissue (pink line) as drawn on the H&E stained slide (**A**). (**B**) Consecutive slide with immunohistochemical CEA staining. (**C**) fluorescence flatbed image of the same slide as (**A**). (**D**) autoradiography image of the same slide as (**A**).

**Figure 3 cancers-12-00987-f003:**
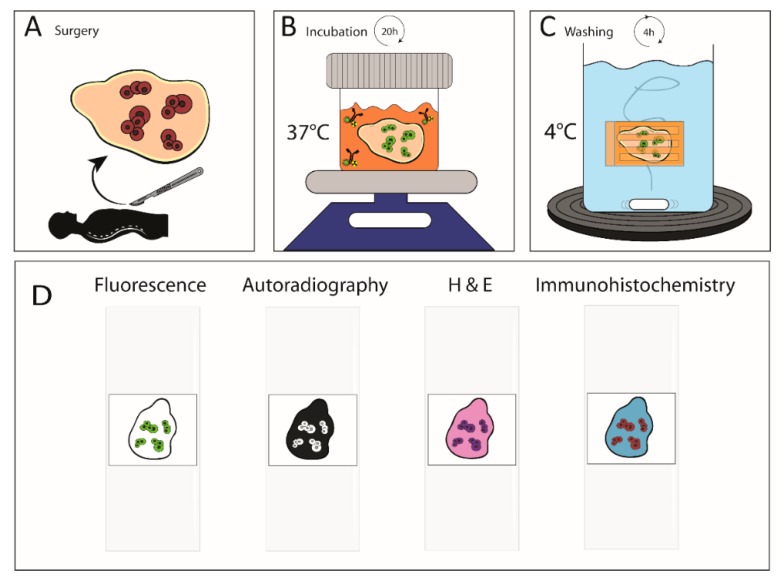
Method of ex vivo incubation of surgical tissue samples with ^111^In-DOTA-hMN-14-IRDye800CW. (**A**) surgical resection of tumor specimen and tissue preparation. (**B**) Overnight incubation with ^111^In-DOTA-hMN-14-IRDye800CW at 37 °C on orbital shaker with antibiotic additives. (**C**) Four hours washing of unbound antibody-conjugate at 4 °C. (**D**) Formalin-fixation, paraffin embedding and mounting of serial slices on glass slides for (immuno)histochemical analyses.

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
