# Peer review of "Ex Vivo Assessment of Tumor-Targeting Fluorescent Tracers for Image-Guided Surgery"

_cancers, 2020, doi:10.3390/cancers12040987_

Round 1

Reviewer 1 Report

This paper describes ex vivo assessment of surgical excised tissue after incubation with a fluorescent or multimodal variant of a CEA-targeting antibody with the aim to determine the tumor-to-background ratio and assess the feasibility of the use of this tracer in a clinical setting. I fully agree with the authors that there is a huge translational gap that needs to be overcome, and that the use of surgical samples could indeed provide an elegant intermediate evaluation step. However, the applied setup does not seem to extend beyond an IHC application, limiting its additional value. Also, incubation times of 20 hours (plus 4 hours washing) are required to obtain a sufficient signal in this case. Although this could be representative for the use of antibodies in vivo (which is also one of their main limitations), in an ex vivo situation this is extremely long. Especially when compared to in vitro staining with such targeted tracers (in cells and tissue), which normally provides excellent staining within minutes. Such long incubation times could also vastly affect the composition of the tissue, and thus possibly, the outcome of the experiment. A further increase of the incubation time, as suggested in the discussion, would therefore not be ideal and could also impose an increase in fluorescence due to unspecific binding. Furthermore, in Figure 2 two clear false positives and a number of questionable false positives can be seen in the conjugate in normal tissue group. The effect of these false-positive values on the applicability of this approach should at least be discussed.

Reviewer 2 Report

In this manuscript, the authors introduced an anti-carcinoembryonic antigen (CEA) antibody with a fluorescent and radioactive label to achieve high tumor-to-background tissue ratios (TBR). This bimodal imaging strategy was used to evaluate the clinical feasibility of the tracer for image-guided surgery. Overall, the design of the material and methods is nice. The delineation of the tumor is remarkable for the ex vivo study on human tissues. However, several suggestions below are made below for improvements.

Main concerns:

1.Few information on the dual-labeled CEA antibody conjugate was provided. For example, (a) no physical characterization was provided to verify the conjugation is successful; (b) how about the toxicity of this biomaterials. 

2 Is the conjugate stable enough to remain intact in the whole process to provide the accurate signal for the tumor imaging?

Minor points:

3 what might be the reason for the difference between the two methods in supplementary figure 1.

4 It looks like the radiolabeling method is not as good as fluorescence method. What is the explanation?

5 The scale bars in Figure 3 and Figure S3b were not clearly labelled

6 In Figure S3, the images were from one patient or representative for several (n=?).

Author Response

Response to reviewer 2 comments:

In this manuscript, the authors introduced an anti-carcinoembryonic antigen (CEA) antibody with a fluorescent and radioactive label to achieve high tumor-to-background tissue ratios (TBR). This bimodal imaging strategy was used to evaluate the clinical feasibility of the tracer for image-guided surgery. Overall, the design of the material and methods is nice. The delineation of the tumor is remarkable for the ex vivo study on human tissues. However, several suggestions below are made below for improvements.

Main concerns:

1.Few information on the dual-labeled CEA antibody conjugate was provided. For example, (a) no physical characterization was provided to verify the conjugation is successful; (b) how about the toxicity of this biomaterials.

Response 1: We would also like to thank reviewer 2 for his or her appreciated critical appraisal of our manuscript.

On question (a): Our research group has extensive experience with dual-labeling of antibodies. The preparation and testing of the antibody was performed  as described by Rijpkema et al. 2014 [1]. We have added a phrase to the methods section to emphasize this, on page 6 line 247-248: “Analyses of the fluorescence signal of the antibody conjugate were performed as described by Rijpkema et al.“ The binding assay confirms the specific binding of the antibody and is described in the methods section 4.5.

Question (b): We have previously shown that the use of the antibody does not induce any toxicity in mice [1, 2]. Labetuzumab, the antibody that is used in the manuscript, has been tested extensively for toxicity in humans [3, 4]. These studies used much higher doses of the antibody than in our study. Since we are interested in the diagnostic use of this conjugated antibody, we use much lower doses and expect no toxicity in patients.

2 Is the conjugate stable enough to remain intact in the whole process to provide the accurate signal for the tumor imaging?

Response 2: The results from the studies mentioned above (reference [1] and[2]) show that the conjugate is stable enough in vivo to provide accurate signal for tumor imaging, even 7 days after intravenous injection in mice. Therefore,  there is no reason to assume that the incubation method deteriorates the antibody conjugate more than in vivo.

Minor points:

3 what might be the reason for the difference between the two methods in supplementary figure 1.

Response 3: Indeed there is a difference in the TBR between fluorescence and autoradiography in supplementary figure 1. The reason for this is that on a microscopic scale the spatial resolution of fluorescence imaging is much higher than in autoradiography.

4 It looks like the radiolabeling method is not as good as fluorescence method. What is the explanation?

Response 4: As mentioned in comment 3, the autoradiography of figure S1 indeed shows a lower TBR than the fluorescence signal TBR. Because the spatial resolution of autoradiography is lower, the contrast between tumor and non-tumor tissue is less clear, leading to lower TBR’s. However, in patients the radiosignal will not be used on a microscopic level, but only on a macroscopic level, for example for whole body imaging, to detect tumor lesions, or to quantify the signal that is present in tissues of interest.

5 The scale bars in Figure 3 and Figure S3b were not clearly labelled

Response 5: We have now adjusted the scale bars of figure 3 and S3a/b, so that they are more clear for the reader.

6 In Figure S3, the images were from one patient or representative for several (n=?).

Response 6: The tumors are representative for all patients. The two specific tumors from the images in figure supplementary 3 are tumors from two different patients. We have added the phrase “(other patient than depicted in figure S3a/b)” to the legend of the figures, to clarify this.

References for response to reviewer 2:

  1. Rijpkema, M., et al., SPECT- and fluorescence image-guided surgery using a dual-labeled carcinoembryonic antigen-targeting antibody. J Nucl Med, 2014. 55(9): p. 1519-24.
  2. Hekman, M.C.H., et al., Detection of Micrometastases Using SPECT/Fluorescence Dual-Modality Imaging in a CEA-Expressing Tumor Model. J Nucl Med, 2017. 58(5): p. 706-710.
  3. Hajjar, G., et al., Phase I radioimmunotherapy trial with iodine-131--labeled humanized MN-14 anti-carcinoembryonic antigen monoclonal antibody in patients with metastatic gastrointestinal and colorectal cancer. Clin Colorectal Cancer, 2002. 2(1): p. 31-42.
  4. Sharkey, R.M., et al., Evaluation of a complementarity-determining region-grafted (humanized) anti-carcinoembryonic antigen monoclonal antibody in preclinical and clinical studies. Cancer Res, 1995. 55(23 Suppl): p. 5935s-5945s.

Round 2

Reviewer 2 Report

My concerns were mostly addressed.

Author Response

Thank you once again for critically reviewing our manuscript.